# Excitatory and inhibitory receptors utilize distinct post- and trans-synaptic mechanisms in vivo

**Taisuke Miyazaki[1,2,3], Megumi Morimoto-Tomita[1], Coralie Berthoux[4], Kotaro Konno[3], Yoav Noam[1], Tokiwa Yamasaki[1], Matthijs Verhage[5], Pablo E Castillo[4], Masahiko Watanabe[3], Susumu Tomita[1]***

[1]Department of Cellular and Molecular Physiology, Department of Neuroscience, Yale University School of Medicine, New Haven, United States; [2]Department of Health Sciences, School of Medicine, Hokkaido University, Sapporo, Japan; [3]Department of Anatomy, Faculty of Medicine, Hokkaido University, Sapporo, Japan; [4]Dominick P. Purpura Department of Neuroscience, Albert Einstein College of Medicine, Bronx, United States; [5]Department of Clinical Genetics, Center for Neurogenomics and Cognitive Research (CNCR), VU University Amsterdam and VU Medical Center, Amsterdam, Netherlands

**\*For correspondence:**
susumu.tomita@yale.edu

**Competing interest:** The authors declare that no competing interests exist.

**Abstract** Ionotropic neurotransmitter receptors at postsynapses mediate fast synaptic transmission upon binding of the neurotransmitter. Post- and trans-synaptic mechanisms through cytosolic, membrane, and secreted proteins have been proposed to localize neurotransmitter receptors at postsynapses. However, it remains unknown which mechanism is crucial to maintain neurotransmitter receptors at postsynapses. In this study, we ablated excitatory or inhibitory neurons in adult mouse brains in a cell-autonomous manner. Unexpectedly, we found that excitatory AMPA receptors remain at the postsynaptic density upon ablation of excitatory presynaptic terminals. In contrast, inhibitory GABA$_A$ receptors required inhibitory presynaptic terminals for their postsynaptic localization. Consistent with this finding, ectopic expression at excitatory presynapses of neurexin-3 alpha, a putative trans-synaptic interactor with the native GABA$_A$ receptor complex, could recruit GABA$_A$ receptors to contacted postsynaptic sites. These results establish distinct mechanisms for the maintenance of excitatory and inhibitory postsynaptic receptors in the mature mammalian brain.

## Introduction

Fast excitatory and inhibitory synaptic transmissions in the mature brain are mostly mediated by ionotropic AMPA-type glutamate receptors (AMPARs) and GABA$_A$ receptors (GABA$_A$Rs), respectively. The maintenance of these receptors at the corresponding postsynapse involves various components, including postsynaptic cytosolic scaffolding proteins and trans-synaptic components, such as membrane and secreted proteins (*Barrera-Ocampo and Chater, 2013*; *Gerrow and El-Husseini, 2007*; *Luscher et al., 2011*; *Martenson and Tomita, 2015*; *Moss and Smart, 2001*). However, it is less clear which of these mechanisms is crucial for maintaining neurotransmitter receptors at postsynapses in vivo.

Acetylcholine receptors remain at synapses of the neuromuscular junction after denervation, highlighting the importance of the postsynaptic machinery for its maintenance at synapses (*Hartzell and Fambrough, 1972*). As denervation of specific neurons is not feasible in the brain, alternative approaches such as X-irradiation have been attempted to induce neuronal loss, for example, in the cerebellum (*Altman and Anderson, 1972*; *Bayer and Altman, 1975*). The specificity of the neuronal

loss was later improved with genetic and pharmacological approaches, such as by expression of a toxin gene or exogenous receptor/enzyme expression combined with agonist administration (*Gray et al., 2010*; *Palmiter et al., 1987*; *Watanabe et al., 1998*; *Yang et al., 2013*). To date, these methods have been limited in their ability to uncover mechanisms of postsynaptic receptor localization mainly due to pleiotropic effects.

Loss of Syntaxin-binding protein 1 (Stxbp1 – also known as Munc18-1) results in massive neurodegeneration (*Verhage et al., 2000*), and expression of Cre recombinase in primary cultured neurons from conditional Stxbp1 homozygous mice (*Stxbp1*^fl/fl) induces neuronal death in a cell-autonomous manner (*Heeroma et al., 2004*). Moreover, Cre recombinase expression under *Pcp2* and *Sert* promoters results in degeneration of cerebellar Purkinje cells (PCs) and serotonin (5-HT) neurons, respectively (*Dudok et al., 2011*; *Heeroma et al., 2004*). Here, we took advantage of the cell-autonomous elimination observed in Stxbp1 homozygous neurons (*Dudok et al., 2011*; *Heeroma et al., 2004*) to investigate the machinery responsible for maintaining neurotransmitter receptors at postsynapses.

## Results

### Selective removal of granule cells and PCs in the mature brain

The cerebellum consists of the molecular layer (ML), the Purkinje cell layer (PCL), the granular layer (GL), and deep cerebellar nuclei (DCN) in the white matter (*Figure 1A*). Granule cells (GCs) in the GL form excitatory glutamatergic synapses with PCs and molecular layer interneurons (MLIs) in the ML. MLIs form inhibitory GABAergic synapses with PCs, which then form inhibitory synapses with DCN (*Figure 1A*). We took advantage of the well-defined neuronal and synaptic organization of the cerebellum to investigate the roles of presynaptic components in maintaining excitatory or inhibitory receptors.

We crossed conditional Stxbp1 mice (*Stxbp1*^fl/fl) with transgenic mouse lines expressing Cre recombinase under the *Gabra6* promoter for GCs (*Fünfschilling and Reichardt, 2002*) or the *Pcp2* promoter for PCs (*Barski et al., 2000*), and obtained *Gabra6-Cre: Stxbp1*^fl/fl (ΔGC) and *Pcp2-Cre: Stxbp1*^fl/fl (ΔPC) mice. We used littermates that did not carry a transgene of Cre recombinase (control) for comparison. ΔGC and ΔPC mice began to show obvious locomotor phenotypes at around P28 (*Figure 1B* and *Videos 1 and 2*).

Nissl staining demonstrated the loss of GCs in ΔGC mice and PCs in ΔPC mice at 2 months of age (*Figure 1C*), but no obvious differences at postnatal days 7–9 (P7–9) (*Figure 1—figure supplement 1*). The total area of midsagittal cerebellar sections was significantly smaller in ΔGC and ΔPC mice than in control mice (*Figure 1C and D*). In ΔGC mice, the thickness of the GL was reduced by 82%, but the PCL was intact. By contrast, the PCL disappeared with no change of the GL thickness in ΔPC mice. We also found a significant reduction in the ML thickness in both mice: 43.4% and 24.9% reduction in ΔGC and ΔPC mice, respectively (*Figure 1D*). The degrees of reduction are comparable to volume fraction of parallel fiber (PF) (*Napper and Harvey, 1988*) and PC dendrites (*Hamodeh et al., 2010*; *Roth and Häusser, 2001*). These results indicate that GCs and PCs were selectively and effectively eliminated in a cell-autonomous manner from the adult ΔGC and ΔPC cerebella.

### Excitatory receptors remain at postsynapses upon presynaptic ablation

We examined whether presynaptic components are required to maintain excitatory receptors by analyzing receptor localization on presynapse-free postsynapses. GCs project PFs to form excitatory glutamatergic synapses onto PC spines and MLI dendritic shafts (*Figure 1A*). In adult ΔGC mice, PF ablation was confirmed by loss of the PF protein vesicular glutamate transporter (VGluT1) (*Miyazaki et al., 2003*) in the ML, except the very surface of the ML and thinned GL (*Figure 2A*). These results indicate that excitatory PF inputs to PCs and MLIs are mostly ablated in adult ΔGC mice.

Given that orphan GluD2 receptors selectively localize at PF–PC postsynapses (*Landsend et al., 1997*), we utilized GluD2 as a marker of PF–PC postsynapses. Punctate signals of GluD2 were observed in both control and ΔGC mice with a 26 % increase of signal intensity in ΔGC mice (*Figure 2B*), which is presumably due to condensation of GluD2 puncta in a thinned ML (*Figure 1C*). Notably, GluD2 signal was stronger at the surface of the ML (*Figure 2B*), where VGluT1-labeled PFs remained (*Figure 2A*). In electron microscopy analysis (*Figure 2C*), all PC spines were contacted and formed asymmetrical synapses in the control mice (contacted spine), whereas most spines in the

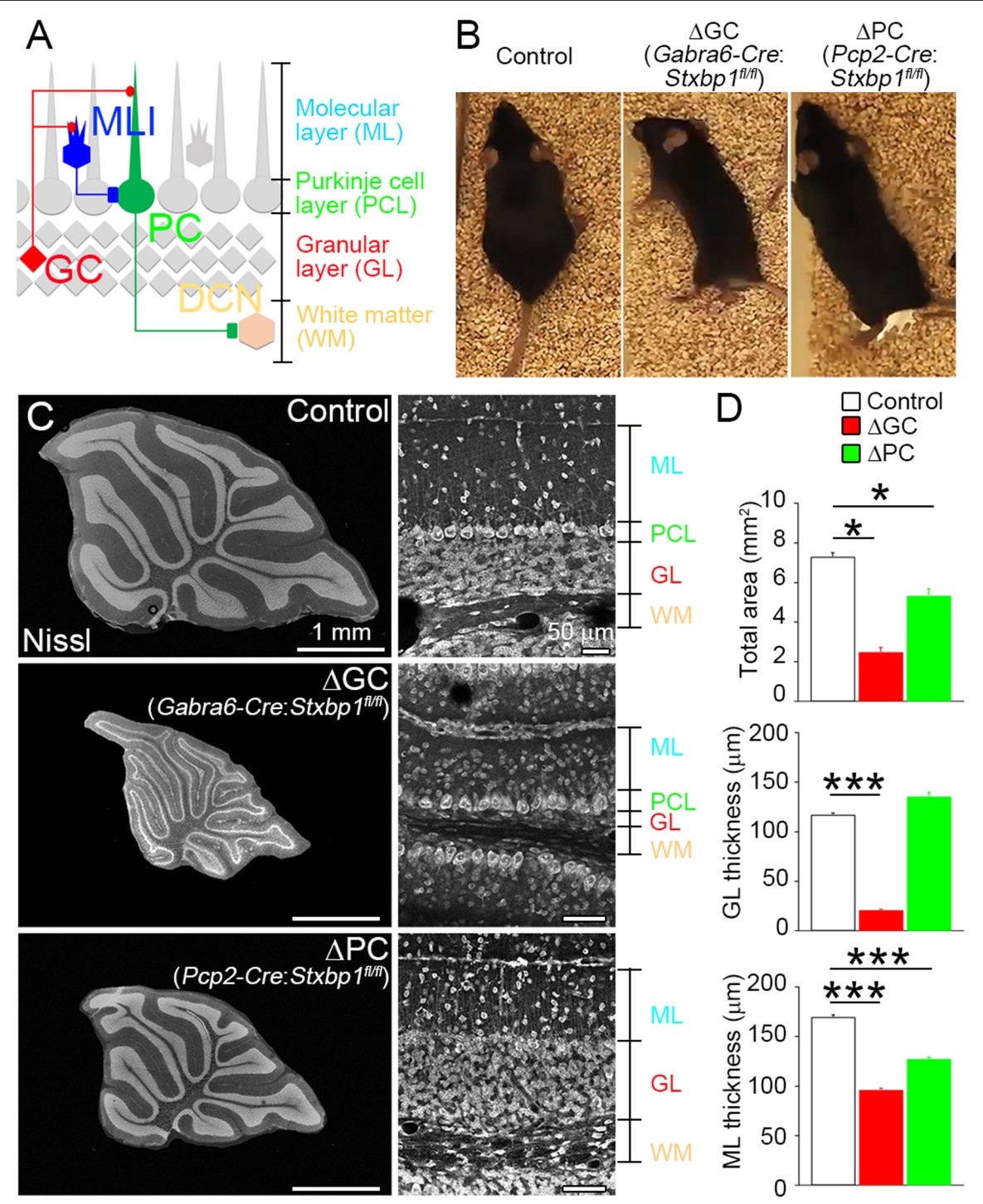

**Figure 1.** Adult cerebellar organization after cell-autonomous elimination of neurons. (**A**) Synaptic circuit organization in the adult cerebellum. Granule cells (GCs) send excitatory inputs to Purkinje cells (PCs) and molecular layer interneurons (MLIs). MLIs form inhibitory synapses on to PCs that send inhibitory inputs to the deep cerebellar nucleus (DCN). (**B**) Cerebellar GC- and PC-specific conditional Stxbp1 homozygous mice (ΔGC mice and ΔPC mice, respectively) manifest ataxia at 5–6 months of age (see Extended Data *Videos 1 and 2*). (**C**) Cerebellar histology of control (top), ΔGC (middle), and ΔPC (bottom) mice at P30. (**D**) Significant decrease in the total cerebellar area and the thicknesses of granular layer (GL) and molecular layer (ML) (*n* = 45–67 regions from three mice each). Note that gross histoarchitecture of cerebellum is maintained in ΔGC and ΔPC mice. Data are means ± SEMs; Kruskal–Wallis test with Steel post-test; *p < 0.05, ***p < 0.001. The numerical values are summarized in source data.

The online version of this article includes the following source data and figure supplement(s) for figure 1:

*Figure 1 continued on next page*

*Figure 1 continued*

**Source data 1.** Adult cerebellar organization after cell-autonomous elimination of neurons.

**Figure supplement 1.** Normal cerebellar histology of mouse brains at P7–9.

ΔGC mice (87.1% ± 1.5%) were free of innervation (free spine) and wrapped by Bergmann glia (BG in *Figure 2C*). Surprisingly, no difference was observed in the spine number between control and ΔGC mice (control, 0.45 ± 0.03 and ΔGC, 0.42 ± 0.02 spines per µm of PC dendrite, *n* = 142 from three mice each). Moreover, thick postsynaptic density (PSD), which is typical of asymmetrical synapses, was also discerned on free spines. The immunogold AMPAR particle density in dendritic spines colabeled with PC protein Car8 was similar between contacted spines in control mice and free spines in ΔGC mice (*Figure 2D*).

We next examined whether AMPARs at free spines were functional by performing two-photon uncaging of glutamate on individual dendritic spines in acute sagittal cerebellar slices perfused with MNI-glutamate (*Figure 2E*). Dendritic spines were identified by loading PCs with Alexa 594 (*Figure 2F*). We found that uncaging glutamate-evoked excitatory postsynaptic currents were undistinguishable in control and ΔGC mice (control, 32 ± 3.9 pA, *n* = 6 from three mice; ΔGC, 30 ± 3.9 pA, *n* = 7 from four mice; p = 0.8037, unpaired *t*-test), and these responses were abolished by the AMPAR antagonist NBQX (*Figure 2G and H*). These results indicate that postsynaptic structure and AMPAR localization at the PSD are maintained on PC spines in the absence of presynaptic terminals (*Figure 2I*).

Unlike spiny PF–PC synapses, PF–MLI synapses are established on the dendritic shaft. In immunostaining for PSD-95, which accumulates at the MLI postsynapse (*Fukaya and Watanabe, 2000*), ΔGC mice exhibited a 46 % reduction in the density of, and a 43 % increase in the size of individual PSD-95 puncta (*Figure 3A*). Electron microscopy showed that all PSDs were observed on the postsynaptic side of axodendritic shaft PF–MLI synapses in control mice (*Figure 3B*, left), while in ΔGC mice the vast majority (85.8% ± 2.2%, *n* = 6 images from three mice each) was found in both sides of atypical dendrodendritic contacts between MLIs (*Figure 3B*, right). At this unique contact site, we did not observe accumulation of synaptic vesicles with around 40 nm in diameter (*Figure 3B*). The cleft width of the dendrodendritic contact in ΔGC mice (30.0 ± 1.0 nm) was wider than that of axodendritic synapse in control mice (15.8 ± 0.5 nm, *n* = 63–73 from three mice each). Notably, while AMPARs were localized only on the postsynaptic side in control mice, they were detected on both sides of dendrodendritic contacts and exhibited a twofold increase in ΔGC mice (*Figure 3C*). Furthermore, presynaptic active zone protein bassoon, which was localized at the PF–MLI synapses in control mice (*Figure 3D and E*, left), disappeared from dendrodendritic contact in ΔGC mice (right). We also confirmed the lack of VGluT1 labeling at around dendrodendritic contact (*Figure 3F*). These results suggest that the loss of presynaptic elements in vivo induces homophilic interaction between excitatory postsynapses on neighboring MLIs and transforms PF–MLI synapse into unique dendrodendritic contact, at which AMPARs continue to localize at the PSD-like specialization, but presynaptic molecules are no longer clustered (*Figure 3G*). Therefore, after presynaptic neuron ablation, AMPAR clustering is maintained at two distinct types of excitatory synapses, both being at the PSD-like specialization on dendritic spines of PCs and on dendritic shafts of MLIs.

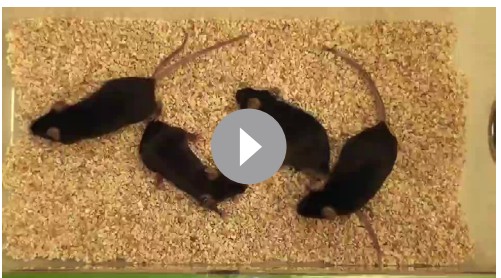

**Video 1.** Sample video of two mice eliminating cerebellar granule cells (ΔGC) and two control mice.
https://elifesciences.org/articles/59613/figures#video1

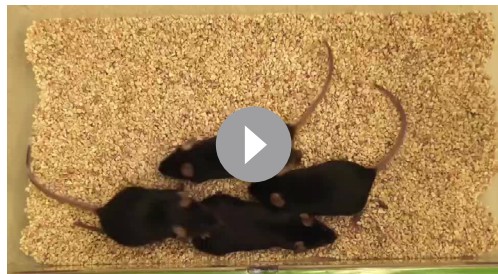

**Video 2.** Sample video of two mice eliminating cerebellar Purkinje cells (ΔPC) and two control mice.
https://elifesciences.org/articles/59613/figures#video2

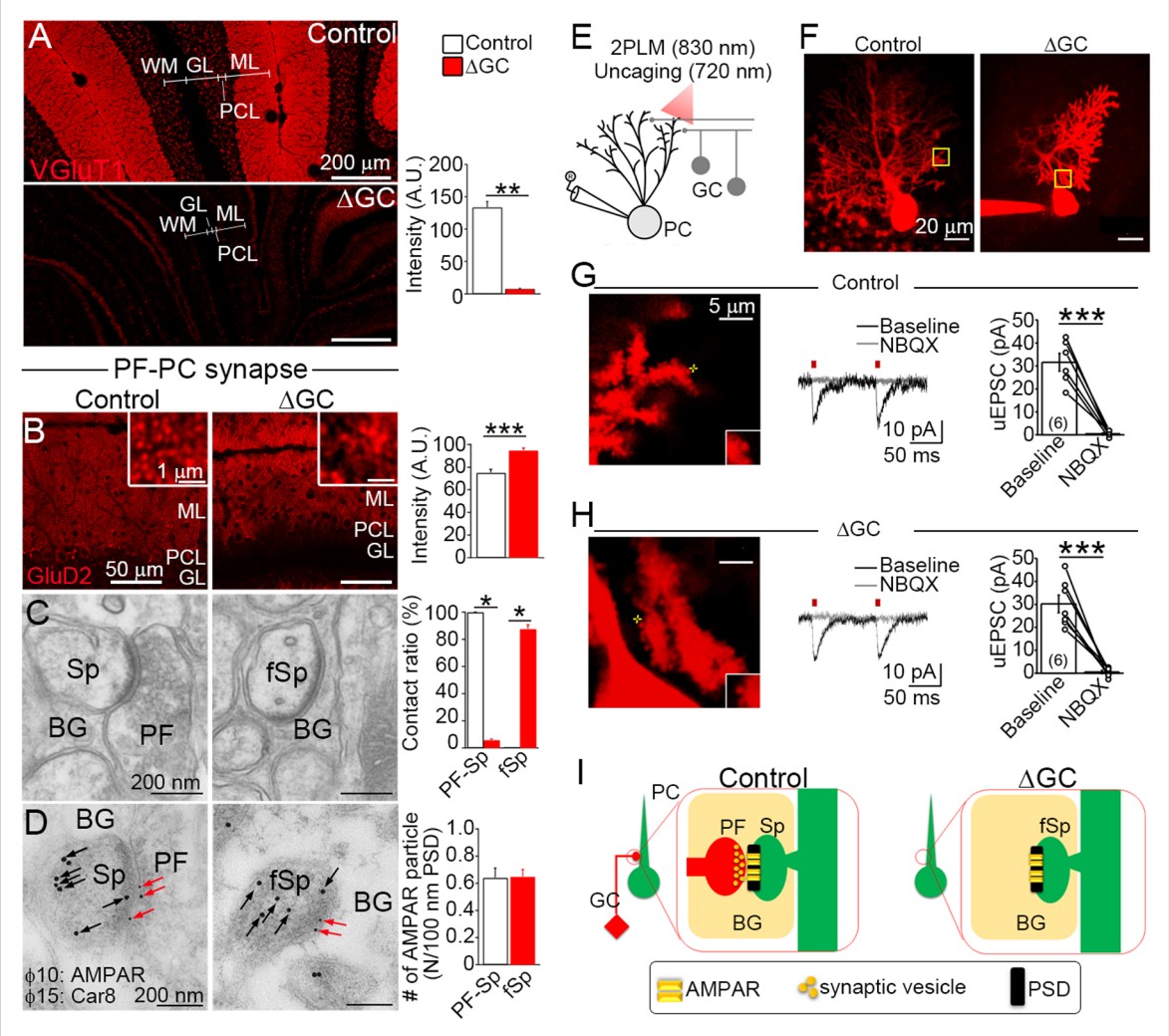

**Figure 2.** Excitatory AMPARs remain at spiny synapses without presynaptic terminals. (**A**) Vesicular glutamate transporter (VGluT1) signal from GC axons, that is, parallel fibers (PFs), is significantly decreased in the molecular layer (ML) of the ΔGC mice (n = 6 images from three mice each). A.U.: arbitrary units. (**B–D**) Analysis of parallel fiber (PF)–Purkinje cell (PC) spiny synapses. (**B**) GluD2 show punctate localization in the ML and increase signal intensity in the ΔGC mice (n = 20 images from three mice each). A.U.: arbitrary units. (**C**) Electron micrographs showing a spine (Sp) making synaptic contact with a PF in a control mouse and a free spine (fSp) lacking synaptic contact in a ΔGC mouse (n = 6 regions from three mice each). BG: Bergmann glia. (**D**) Postembedding immunogold for panAMPAR (10 nm, red arrows) on Car8 (15 nm, black arrows)-labeled PC spines (Sp) or a free spine (fSp) in control and ΔGC mice, respectively (n = 114–130 from three mice each). BG: Bergmann glia. (**E**) Schematic diagram illustrating the experimental design of combined electrophysiology, imaging and uncaging in acute cerebellar slices from ΔGC and control littermates. Excitation wavelengths employed were 830 nm (for imaging) and 720 nm (for glutamate uncaging). (**F**) 2 P reconstruction of a Purkinje cells loaded with Alexa 594 in ΔGC and control mice. Left, example of the position of the uncaging laser beam (yellow cross) in control (**G**) and ΔGC (**H**) littermates. Middle, representative uncaging glutamate-evoked excitatory postsynaptic currents (uEPSCs). NBQX (10 μM) was bath applied to block AMPARs. Right, summary plot demonstrating that NBQX (10 μM) inhibits uEPSC evoked by glutamate uncaging (control, 6 cells from 3 mice; ΔGC, 7 cells from 4 mice). Data are presented as mean ± SEM. ***p < 0.001, paired *t*-test. (**I**) After ablation of PFs AMPAR clustering and postsynaptic density (PSD)-like specialization are maintained on free PC spines in ΔGC mice. BG: Bergmann glia. Data are means ± SEMs, Mann–Whitney *U*-test (**A–D**), *p < 0.05, **p < 0.01, ***p < 0.001. The numerical values are summarized in source data.

The online version of this article includes the following source data for figure 2:

**Source data 1.** Excitatory AMPARs remain at spiny synapses without presynaptic terminals.

## Inhibitory postsynaptic receptor localization requires presynaptic components

We next evaluated the role of presynaptic components in maintaining inhibitory receptors at GABAergic PC–DCN synapses (*Figure 1A*). In ΔPC mice, calbindin-positive PC axon terminals and

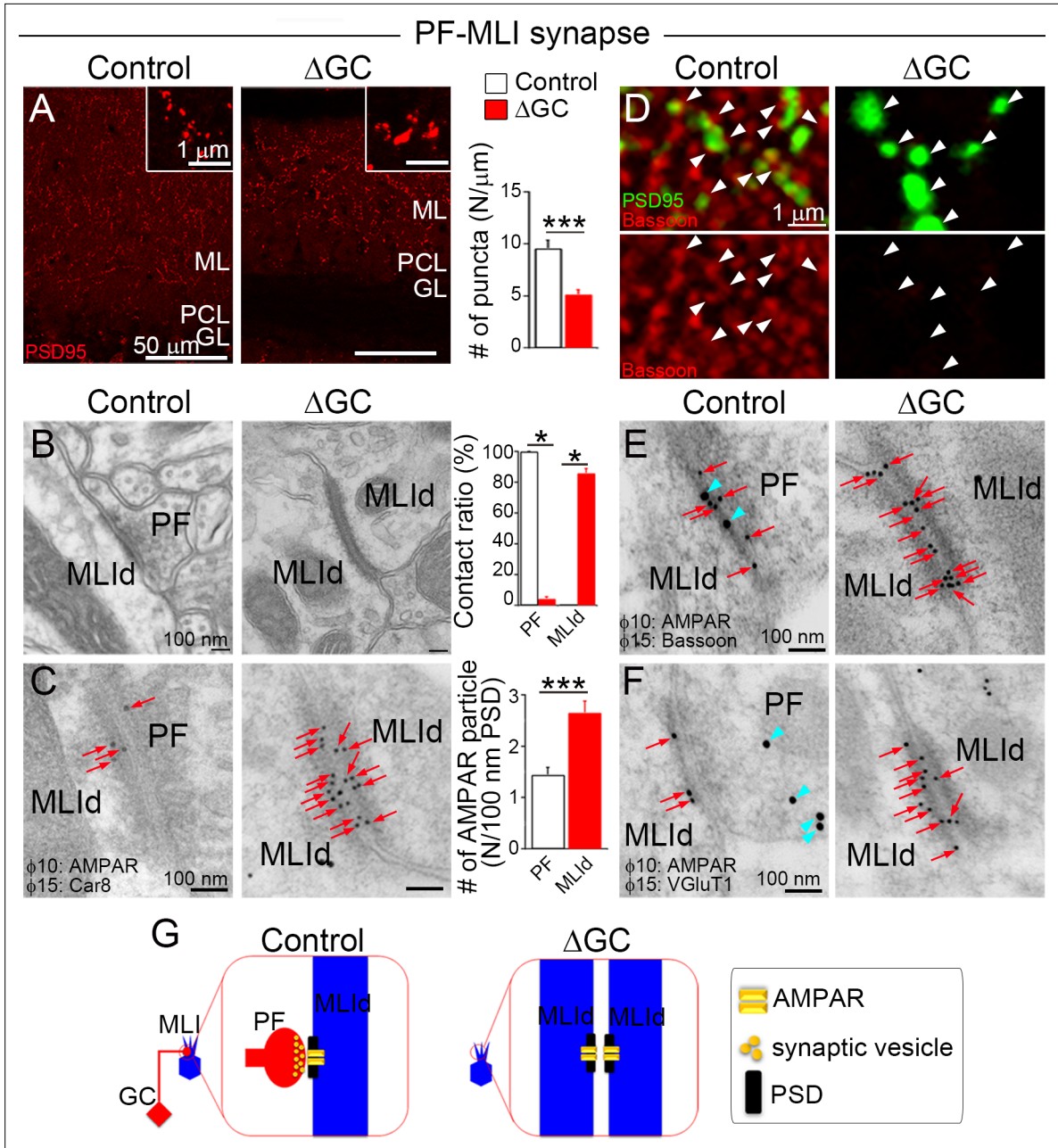

**Figure 3.** Excitatory AMPARs remain at shaft synapses without presynaptic terminals. (**A**) The number of postsynaptic density (PSD)-95 puncta is reduced (*n* = 6 images from three mice each) and individual puncta is enlarged in molecular layer (ML) of the ΔGC mice (*n* = 1421 puncta in control and 686 in ΔGC from three mice each). (**B**) Electron micrographs showing a typical parallel fiber (PF)–molecular layer interneuron (MLI) synapse on MLI dendritic shaft (MLId) shaft synapse in a control mouse and an atypical dendrodendritic contact between MLIs in a ΔGC mouse (*n* = 6 images from three mice each). (**C**) Postembedding immunogold for panAMPAR (10 nm, red arrows) and Car8 (15 nm) (*n* = 25–42 from three mice each). Dendrites of MLI are identified by negative labeling for Car8 and the lack of dendritic spines. A bar graph shows the contact ratio of MLId with PF terminals or other MLId. (**D**) Double immunofluorescence for bassoon (red) and PSD-95 (green). Arrowheads indicate PSD-positive puncta. Note that PSD-95 signals are facing to bassoon signals in control. (**E, F**) Double-labeling postembedding immunogold for panAMPAR (10 nm, red arrows) (*n* = 25–42 from three mice each) and active zone protein, bassoon (15 nm, blue arrowheads in E) or synaptic vesicle-associated protein vesicular glutamate transporter (VGluT1) (15 nm, blue arrowheads in F). Presynaptic proteins bassoon and VGluT1 are not detected at and around dendrodendritic contact sites. (**G**) After ablation of PFs, AMPAR clustering and PSD-like specialization are maintained in dendrodendritic contact sites between MLIs in ΔGC mice. Data are means ± SEMs, Mann–Whitney *U*-test (**A–C**), *p < 0.05, ***p < 0.001. The numerical values are summarized in source data.

The online version of this article includes the following source data for figure 3:

**Source data 1.** Excitatory AMPARs remain at shaft synapses without presynaptic terminals.

vesicular inhibitory amino acid transporter (VIAAT)-positive inhibitory terminals were substantially reduced in the DCN (*Figure 4A*). A few VIAAT clusters remained in the DCN of ΔPC mice, which likely originate from local inhibitory DCN neurons (*Uusisaari and Knöpfel, 2011*), residual 18.9 % PCs, or both.

In control mice, the GABA$_A$Rα1 subunit was clustered on somatic membranes of DCN neurons just underneath VIAAT-positive terminals (*Figure 4B and C*). The number of GABA$_A$Rα1 puncta was significantly reduced in ΔPC mice (*Figure 4C and D*), corresponding to the reduction in VIAAT puncta (*Figure 4D*, bottom). On the DCN somata, the remaining GABA$_A$Rα1 signal was either dispersed weakly along the somatic membrane or accumulated intensely at sites just underneath remaining VIAAT puncta (*Figure 4C*). These results suggest presynaptic requirement for the maintenance of inhibitory GABA$_A$R clustering at PC–DCN synapses (*Figure 4E*).

To further investigate the role of presynaptic components in inhibitory receptor localization at GABAergic MLI–PC synapses, we adopted an adenoassociated virus (AAV)-based approach (*Figure 5A*), because of lethality of *Stxbp1*$^{fl/fl}$ mice when crossed with Cre lines (Nos1-Cre and PV-Cre) for MLI ablation. As candidate promoters in AAV vectors expressing GFP(green fluorescent protein), we cloned 2 kb genomic fragments upstream of the TATA boxes from three genes, *Nos1*, *Grin3a*, and *Kit*, as they are explicitly expressed in adult MLIs according to the Allen Brain Atlas (https:// portal.brain-map.org/). Among these constructs, we found that only the *Nos1* promoter showed GFP expression in MLIs locating at lower ML, but not in PCs (*Figure 5B and C*). Next, we introduced the *Nos1* promoter-driven Cre recombinase into *Stxbp1*$^{fl/fl}$ cerebella. By immunostaining for parvalbumin (PV) and VIAAT, mice expressing *Nos1* promoter-driven Cre recombinase showed a significant reduction in neuronal elements of lower MLIs or basket cells (ΔMLI), that is, PV-labeled somata in the lower ML VIAAT-labeled presynaptic terminals around PC somata, and PV/VIAAT-labeled pinceau formation at the base of PC somata (*Figure 5D–F*). Importantly, GABA$_A$Rα1 clusters on PC somata were also reduced significantly in ΔMLI, especially at somatic surface devoid of contact with VIAAT-labeled presynaptic terminals (*Figure 5E–H*). Therefore, after presynaptic ablation, GABA$_A$R is declustered at two distinct inhibitory cerebellar synapses.

## GABA$_A$Rs can be recruited trans-synaptically to neighboring postsynaptic sites

Our results suggest that presynaptic components trans-synaptically control postsynaptic clustering of GABA$_A$R in vivo. If so, ectopic expression of presynaptic regulatory molecules could recruit GABA$_A$Rs to apposed postsynaptic sites. We previously identified five components of the native GABA$_A$R complex, namely GABA$_A$Rα, β, and γ subunits, GARLH3/4, and neuroligin-2 (*Yamasaki et al., 2017*). Knockdown or knockout of GARLH4/LHFPL4 results in a loss of synaptic GABA$_A$R clustering and inhibitory transmission, though GABA$_A$Rs are retained on the neuronal surface (*Davenport et al., 2017*; *Wu et al., 2018*; *Yamasaki et al., 2017*), similar to what we observed in DCN neurons of ΔPC mice (*Figure 4C*). Neuroligin interacts with presynaptic neurexins (*Craig and Kang, 2007*; *Dean and Dresbach, 2006*; *Südhof, 2008*), which are necessary for excitatory or inhibitory transmission (*Chen et al., 2017*). However, it remains unknown whether ectopic neurexin expression at excitatory terminals can trans-synaptically recruit inhibitory GABARs to postsynaptic sites in the brain.

We focused on neurexin-3 (Nrxn3) and excitatory climbing fiber (CF–PC synapses), because *Nrxn3* knockout specifically eliminates inhibitory transmission in the olfactory bulb (*Aoto et al., 2015*), and also because the inferior olivary nucleus (ION) projecting CFs are located distantly from PCs and this provides experimental advantage to avoid potential contamination of AAVs into PCs (*Figure 6A*). We first examined whether neurexin-3 alpha (Nrxn3α) can interact with the GABA$_A$R/GARLH/NL complex. The extracellular domain of Nrxn2β fused with the Fc domain of human immunoglobulin is shown to interact with GABA$_A$Rs (*Zhang et al., 2010*). We purified the extracellular domain of mouse Nrxn3α with splice site four fused with Fc and secreted Fc alone using transfected 293T cells (Nrxn3α-Fc). We found that Nrxn3α-Fc pulled down NL2, GABA$_A$Rγ2 subunits, and GARLH4, but not GluA1 AMPAR, from adult mouse cerebellar lysate, in a calcium-dependent manner (*Figure 6B*).

We next examined whether ectopically expressed Nrxn3α in the excitatory CF terminal can recruit GABA$_A$Rs at the contact site on PCs. We generated AAVs carrying either GFP or Nrxn3α fused with GFP (Nrxn3α-GFP) and injected them into the ION of wild-type mice at P21 (*Figure 6A*). One month after AAV injection, the GFP-positive varicosities were overlapped with CF presynaptic marker VGluT2

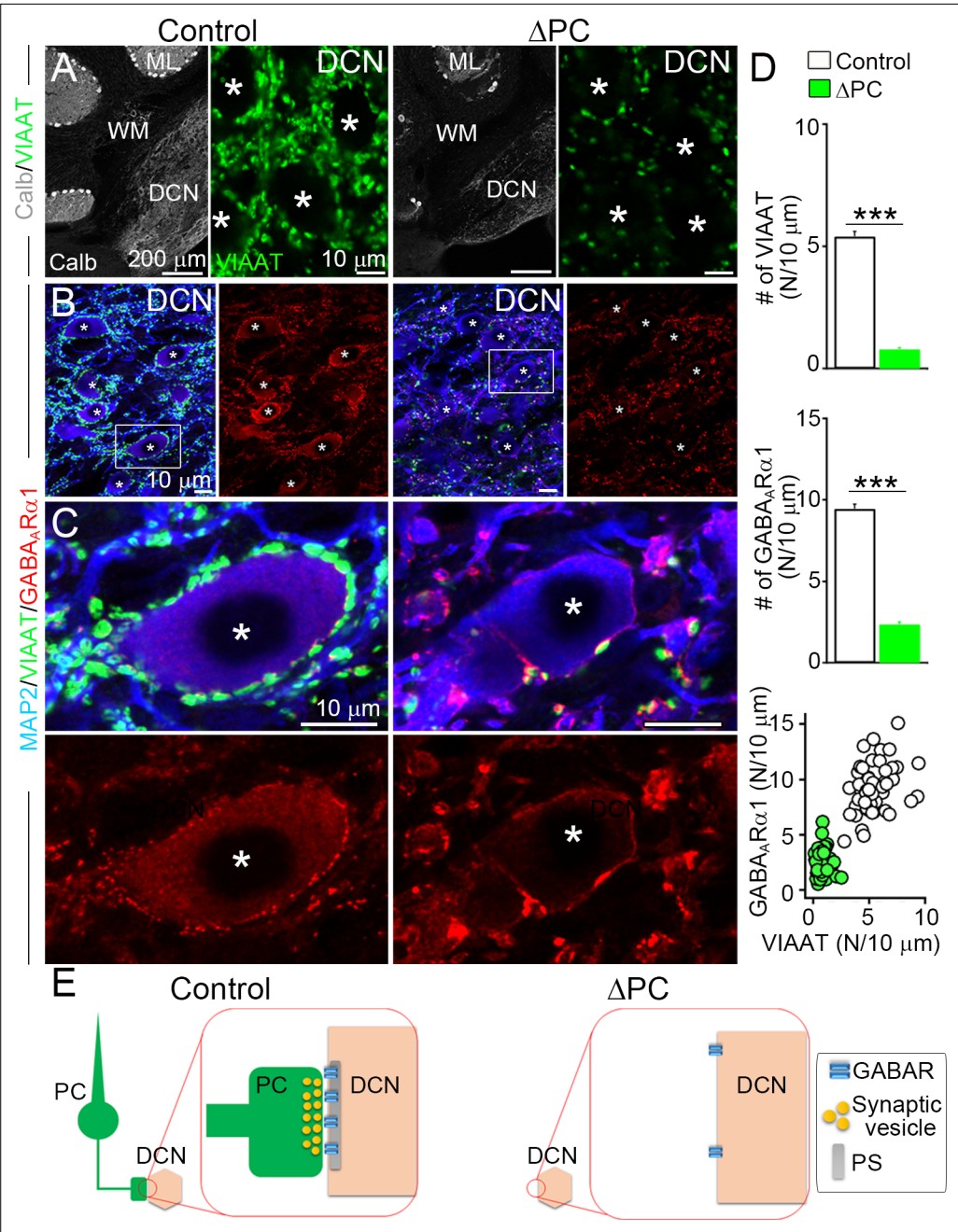

**Figure 4.** Inhibitory GABA$_A$Rs require presynaptic terminals in the deep cerebellar nuclei (DCN). (**A**) Immunofluorescence showing marked reductions of calbindin (Calb)-labeled Purkinje cells (PCs) in the cerebellar cortex and of vesicular inhibitory amino acid transporter (VIAAT)-labeled inhibitory terminals in the DCN of the ΔPC mice. (**B, C**) Triple immunofluorescence for MAP2(microtubule associated protein 2) (blue), VIAAT (green), and GABA$_A$Rα1 (red) in DCN boxed regions in B are enlarged in C. Asterisks indicate cell bodies of DCN neurons, which are fringed by numerous bright clusters of GABA$_A$Rα1 in control mice but are greatly reduced in ΔPC mice. (**D**) Bar graphs showing reduced densities of VIAAT (top)- and GABA$_A$Rα1 (middle)-positive puncta on the surface of DCN neurons in ΔPC mice (number per 10 μm, $n$ = 49–60 neurons from three mice each). Kruskal–Wallis test followed by Steel–Dwass post-test for the comparison of scatter plot (bottom) indicates significant reduction (p < 0.001). (**E**) After ablation of PC terminals, postsynaptic GABA$_A$Rα1 is declustered from somatic membrane of DCN neuron. Data are shown as means ± SEMs, ***p < 0.001. The numerical values are summarized in source data.

The online version of this article includes the following source data for figure 4:

**Source data 1.** Inhibitory GABA$_A$Rs require presynaptic terminals in the deep cerebellar nuclei (DCN).

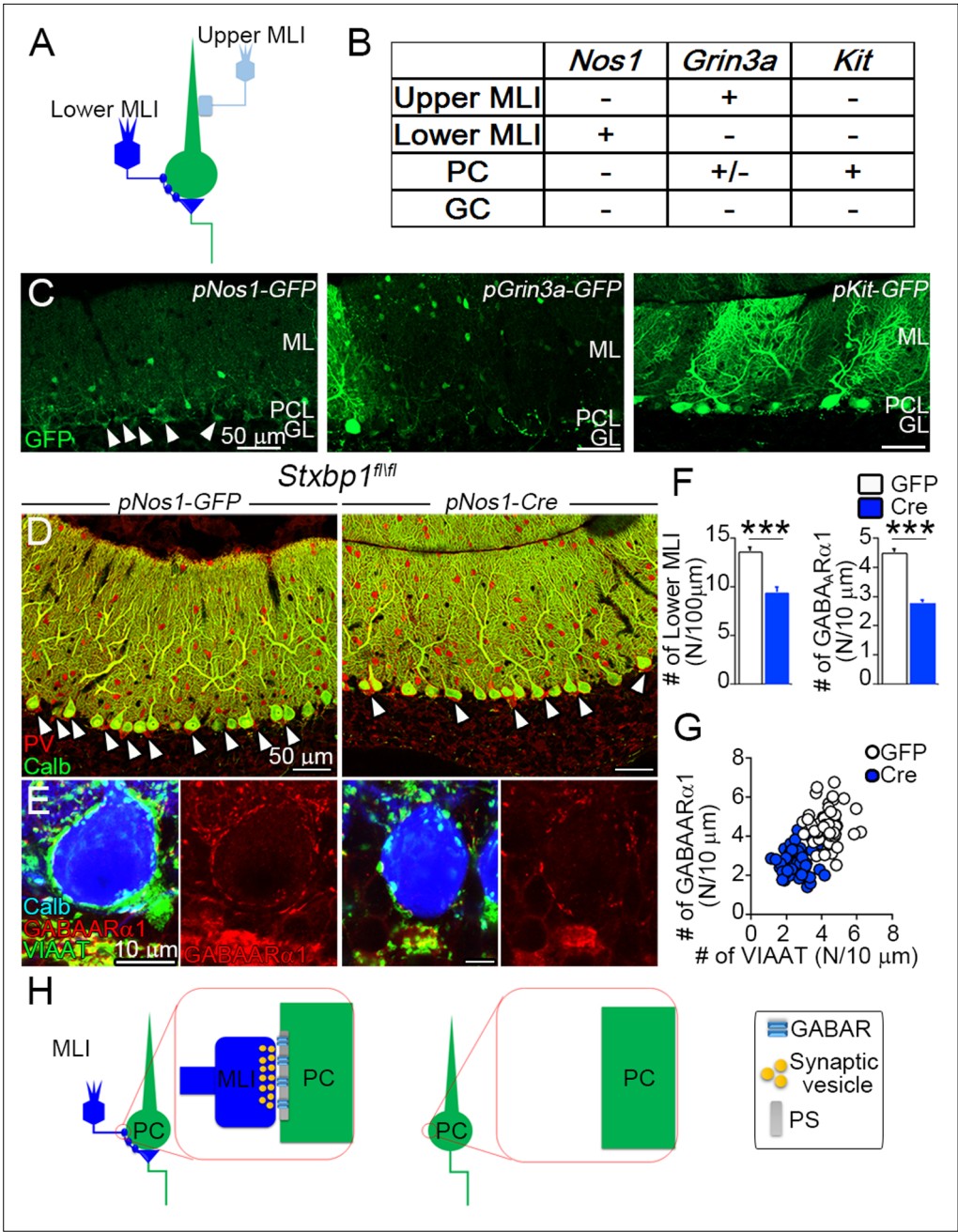

**Figure 5.** Inhibitory postsynaptic sites at Purkinje cell (PC) somas require presynaptic terminals. (**A**) Wiring diagram between molecular layer interneurons (MLIs) and PCs. Lower MLIs, also known as basket cells, innervate PC somata and surround the axon initial segment of PCs with the pinceau formation. Upper MLIs corresponding to stellate cells innervate dendritic shafts of PCs. (**B, C**) Adenoassociated viruses (AAVs) under different promoters targeting to MLIs. Summary of GFP expression is shown in a table (**B**) and representative images (**C**). GFP expression is found preferentially in lower MLIs and pinceau formation in AAV pNos1-GFP (arrowheads in C). (**D, E**) Cerebella of *Stxbp1*^fl/fl mice-injected AAV pNos1-GFP (right) and -Cre (left). Multiple labeling for calbindin (green) and parvalbumin (red) in D and for GABA_AR$\alpha$1 (red), vesicular inhibitory amino acid transporter (VIAAT) (green), and calbindin (blue) in E. (**F**) The densities of lower MLIs (left, N/100 μm of PC layer, $n$ = 12–15 images from three mice each, Mann–Whitney $U$-test) and GABA_AR$\alpha$1-positive puncta on PC somata (right, N/10 μm of membrane, $n$ = 55 neurons from three mice each, Student's $t$-test). (**G**) Scatter plot of the number of GABAAR$\alpha$1- (vertical axis) against VIAAT-positive puncta (horizontal axis) on the surface of each PC soma (N/10 μm, right, $n$ = 55 neuron from 3 mice each, one-way analysis of variance) (ANOVA) ($F$(3,216) = 94.8, with Bonferroni post hoc test, p < 0.001). (**H**) Postsynaptic GABA_AR$\alpha$1 requires inhibitory presynaptic terminals at MLI–PC synapse. After ablation of MLI

*Figure 5 continued on next page*

*Figure 5 continued*

terminals, postsynaptic GABA$_A$Rα1 is declustered from somatic membrane of PC. Data are shown as means ± SEMs; ***p < 0.001. The numerical values are summarized in source data.

The online version of this article includes the following source data for figure 5:

**Source data 1.** Inhibitory postsynaptic sites at Purkinje cell (PC) somas require presynaptic terminals.

and the numbers of GFP-positive varicosities were similar between mice expressing GFP and those expressing Nrxn3α-GFP (*Figure 6C*). These results demonstrate successful introduction of GFP and Nrxn3α-GFP into CFs without affecting target innervation. We next investigated the localization of the GABA$_A$Rα1 subunit, a major GABA$_A$R subunit in PCs (*Fritschy et al., 2006*) on PCs of AAV-injected mice. Some GABA$_A$Rα1-positive puncta overlapped with GFP or Nrxn3α-GFP signals (*Figure 6D and E*). The area and number of varicosities with overlapping GFP and GABA$_A$Rα1 signals were significantly higher in animals injected with Nrxn3α-GFP than in those receiving the vector with GFP alone (*Figure 6E*). These results indicate that ectopic expression of Nrxn3α-GFP in excitatory presynaptic CFs can trans-synaptically recruit inhibitory GABA$_A$Rs to postsynapses on PCs (*Figure 6F*).

## Discussion

We have identified distinct presynaptic dependency of postsynaptic receptors at excitatory and inhibitory synapses in vivo. Specifically, we found that while excitatory AMPARs can localize to PSDs in the absence of presynaptic terminals, inhibitory GABA$_A$Rs require inhibitory presynaptic components for postsynaptic clustering. Altogether, our findings reveal that fundamentally dissimilar machineries maintain different classes of postsynaptic receptors in the mature brain.

### Cell-autonomous elimination of neurons

In this study, we utilized Stxbp1 cKO mice as a powerful tool to eliminate neurons in a cell-autonomous manner (*Figure 1*). Stxbp1 deletion in 5-HT neurons results in early postnatal lethality (*Dudok et al., 2011*). Consistent with this observation, we found neonatal or early postnatal lethality of *Stxbp1*$^{fl/fl}$ mice with various Cre lines, including Emx1-Cre, CaMKII-Cre, Nos1-Cre, Grm2-Cre, Grik4-Cre, and PV-Cre. This lethality precluded the use of these mice in studies of mature synapses. Thus, we utilized injections of AAV carrying Cre recombinase to target-specific neuronal populations (*Figure 5*). By combining the *Stxbp1*$^{fl/fl}$ mouse with the AAV approach, the elimination of specific neurons subtypes can be achieved to evaluate the roles of those neurons on synaptic maintenance, protein function, and behavior, as well as the specificity of Cre recombinase expression. Using this highly selective approach, the overall organization and integrity of the cerebellum remained remarkably intact (*Figure 1*), despite being reduced by >50% in size and the almost complete removal of specific cell populations. Thus, the brain's capacity to remove cell debris and maintain integrity of the remaining network likely copes with such massive cell loss without triggering generalized degenerative responses.

### Machinery for stabilizing receptors at postsynaptic densities

Previous studies proposed that networks of cytosolic, transmembrane, and secreted proteins keep neurotransmitter receptors at postsynaptic sites (*Barrera-Ocampo and Chater, 2013*; *Gerrow and El-Husseini, 2007*; *Luscher et al., 2011*; *Martenson and Tomita, 2015*; *Moss and Smart, 2001*). Surprisingly, our results indicate that presynaptic terminals are required to retain postsynaptic GABA$_A$Rs but not AMPARs (*Figures 2–5*), indicating that presynaptic protein, presumably a secreted or transmembrane protein(s), is required to maintain GABA$_A$Rs at inhibitory synapses, whereas AMPARs require some postsynaptic protein(s), presumably, cytosolic or transmembrane protein(s), for maintenance at excitatory synapses, such as PSD-95-like MAGUK family proteins among others (*Garner et al., 2000*; *Kim and Sheng, 2004*). Also, presynaptic proteins, including Nrxn and Punctin/Madd-4, can induce or specify inhibitory synapses (*Graf et al., 2004*; *Maro et al., 2015*; *Pinan-Lucarré et al., 2014*), which is consistent with our finding that expression of Nrxn3α by excitatory presynaptic neurons recruited GABA$_A$Rs to postsynaptic sites (*Figure 6*). However, not all CF terminals expressing neurexin colocalized with GABA$_A$Rs on PCs (*Figure 6*), and this appeared to be independent of the expression level of Nrxn3α. This could be due to the lack of neurexin at the precise presynaptic

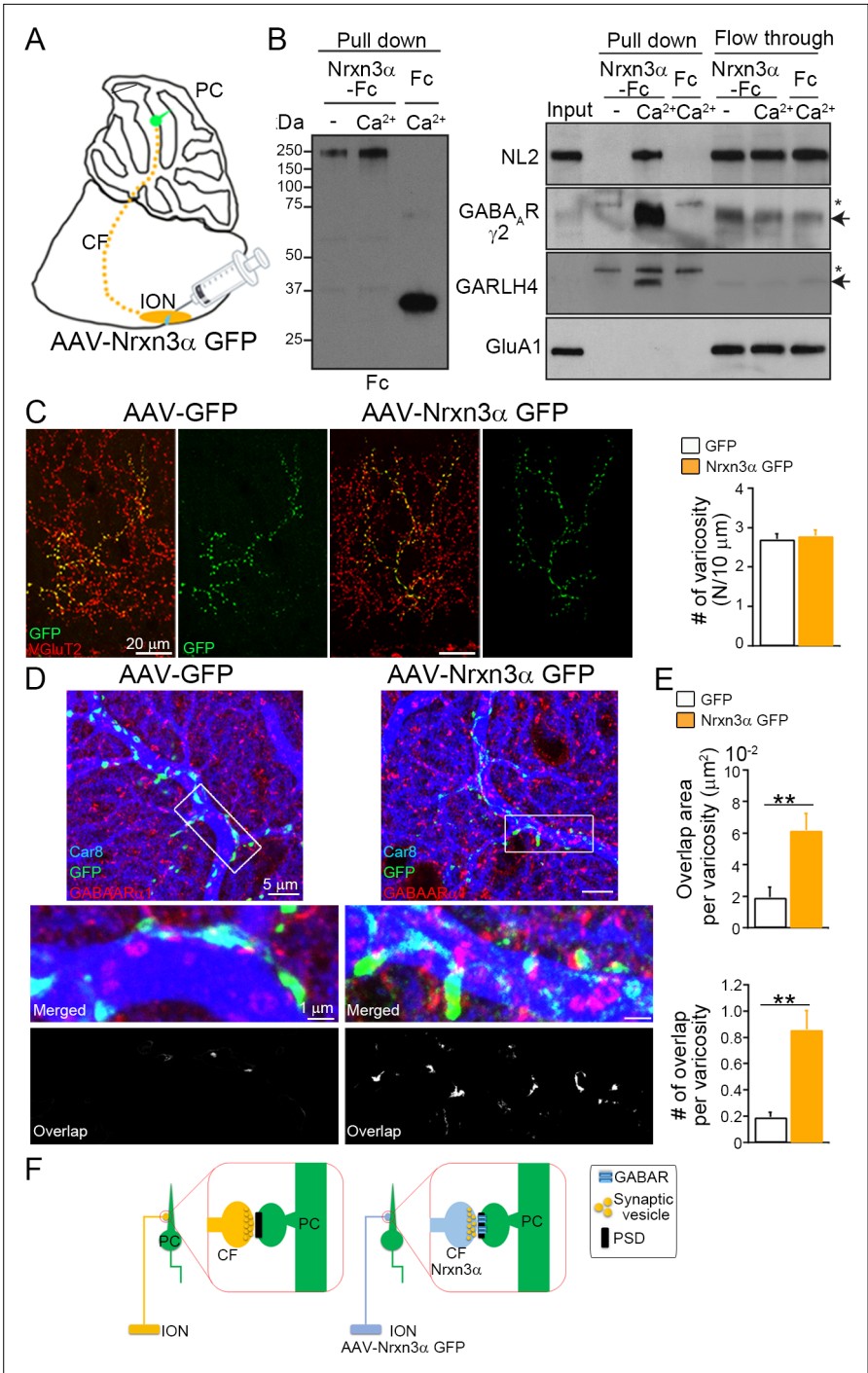

**Figure 6.** Ectopic Nrxn3 expression in excitatory climbing fibers (CFs) recruits inhibitory GABARs. (**A**) Adenoassociated viruses (AAVs) carrying GFP or Nrxn3α fused with GFP (Nrxn3α GFP) are injected into the inferior olivary nucleus (IONs) that project excitatory CFs to Purkinje cells (PCs). (**B**) The extracellular domain of neurexin-3 alpha (Nrxn3α) fused with Fc and secreted Fc are pulled down with protein A and used for pull down with adult mouse cerebellar lysate with and without calcium ($Ca^{2+}$). Nrxn3α-Fc pulls down neuroligin 2 (NL2), GABA$_A$Rγ2, and GARLH4, but not GluA1, in a calcium-dependent manner. The unbound proteins are detected in both conditions with and without calcium, suggesting no obvious protein degradation. Arrows and asterisks indicate specific and nonspecific bands, respectively. The raw images are provided in source data. (**C**) Both GFP and Nrxn3α-GFP (green) are colocalized with presynaptic VGluT2 (red) along CF arbors. No significant difference is found between AAV-GFP and AAV-Nrxn3α GFP in the number of GFP-positive varicosities per 10 μm axon of labeled CFs

*Figure 6 continued on next page*

*Figure 6 continued*

(N/10 µm) (*n* = 34–50 images from three GFP- and Nrxn3α GFP-injected mice). (**D, E**) GABA$_A$R clustering at CF–PC synapses is frequently detected after transfection of AAV-Nrxn3α GFP, as shown by triple immunofluorescence for Car8 (blue), GFP (green), and GABA$_A$Rα1 (red). (**E**) The area and number of overlapped regions between GFP and GABA$_A$Rα1 per single CF varicosity (*n* = 7–18 images from three AAV-injected mice). (**F**) Nrxn3α expression in excitatory CF induces clustering of inhibitory GABA$_A$Rα1 at the CF–PC postsynapse. Data are shown as means ± SEMs, Mann–Whitney *U*-test (**D, E**); **p < 0.01. The numerical values are summarized in source data.

The online version of this article includes the following source data for figure 6:

**Source data 1.** Ectopic Nrxn3 expression in excitatory climbing fibers (CFs) recruits inhibitory GABARs.

nanodomain, as recent studies propose a relationship between the nanodomain structure of synapses and receptor localization (*Biederer et al., 2017*; *Crosby et al., 2019*). Future investigation is needed to characterize receptor and terminal organization in a more quantitative manner.

Excitatory presynaptic proteins have been proposed to be necessary for maintaining AMPARs at synapses (*Barrera-Ocampo and Chater, 2013*; *Gerrow and El-Husseini, 2007*; *Martenson and Tomita, 2015*), whereas we show that the presynaptic terminal is not required for maintaining AMPARs at the synapse (*Figures 2 and 3*). We propose three potential reasons for this discrepancy. (1) Changes in synaptogenesis or AMPAR maintenance at synapses. If proteins are involved in early synaptogenesis, net results from altered synaptogenesis may affect receptor maintenance at postsynaptic sites. In our study, we eliminated presynaptic neurons during or after synapse establishment (*Figure 1* and *Figure 1—figure supplement 1*). (2) Poor resolution in the analysis of synapses. However, we utilized electron microscopy, a high-resolution cell biological approach, and uncaged glutamate responses at single dendritic spines to demonstrate no changes in excitatory receptor localization and function. Because presynaptic neurons were eliminated, we could not evaluate synaptic transmission. Thus, there could be a modest difference in postsynaptic localization of receptors, as a result of variability in image analyses. (3) Divergent mechanisms for maintenance of AMPARs at different type of synapses. However, we showed that trans- and postsynaptic mechanisms for maintenance of AMPARs are conserved between two different types of cerebellar synapses in vivo. Of note, demonstrating this mechanism at other excitatory synapse in the brain requires the complete removal of presynaptic or postsynaptic cell type, which is not easy to achieve in most brain areas.

## Homo- and heterophilic machinery for maintaining synapses

We were able to eliminate excitatory inputs onto PCs and MLIs in vivo, and we observed two unusual types of structures in wild-type neurons, namely, free spines and dendrodendritic postsynaptic contacts, whereas the structure of spines and the PSD were maintained (*Figures 2 and 3*). PCs can generate and maintain spines without presynaptic terminals, but dynamic properties of PC spines might be different without presynapses. Remarkably, Cbln1 KO and GluD2 KO mice showed 80% and 60% of free spines, respectively (*Hirai et al., 2005*; *Kurihara et al., 1997*). It remains unclear why some spines are normal, and some spines lost presynaptic terminals. In addition, free spines have not been observed in different brain regions, for example, hippocampus (*Yuste and Bonhoeffer, 2004*), which may indicate that free spines are unstable in the hippocampus.

The elimination of PF input resulted in the formation of MLI dendrodendritic contact sites (*Figure 3F*). These contacts were unique in that AMPARs were localized at both sides of the contact, with the wider 'cleft' resembling an adhesive junction. It is possible that postsynaptic sites on MLI's dendritic shafts express a protein that can form homophilic interactions, such as between cadherin and SynCAM proteins (*Biederer et al., 2002*; *Gavilondo et al., 1982*). Homophilic interactions may not occur in the presence of higher affinity pre- and postsynaptic protein interactions at normal synapses. In contrast, such homophilic interactions were rare for free spines on PCs in the ΔGC mice, suggesting differences in the molecular compositions of the PSD between PC spines and MLI dendrites or surrounded cell types. For example, after PFs were eliminated, the PSDs of PC spines may have been stabilized by Bergmann glia, specialized astrocytes in the cerebellum that enwrap PCs but not MLIs (*Lemeky-Johnston and Larramendi, 1968*; *Špaček, 1985*). Future identification of molecules necessary to maintain pre- and postsynaptic sites may explain the difference *Continued on next page* in presynaptic-dependent morphology between spine and shaft synapses.

# Materials and methods

## Key resources table

| Reagent type (species) or resource | Designation | Source or reference | Identifiers | Additional information |
|---|---|---|---|---|
| Strain, strain background (*Escherichia coli*) | Putative *Nos1* promoter | BACPAC Resources Center (BPRC) | BAC: RP24-84E3 | |
| Strain, strain background (*Escherichia coli*) | Putative *Grin3a* promoter | BACPAC Resources Center (BPRC) | BAC: RP23-104D24 | |
| Strain, strain background (*Escherichia coli*) | Putative *Kit* promoter | BACPAC Resources Center (BPRC) | BAC: RP23-232H18 | |
| Strain, strain background (*Mus musculus*) | Wild-type (C57BL/6 J) | The Jackson Laboratory | Stock# 00064 | |
| Strain, strain background (*Mus musculus*) | Mouse: *Stxbp1*^lox/lox | **Verhage et al., 2000** | N/A | |
| Strain, strain background (*Mus musculus*) | Mouse: Tg(*Gabra6-Cre*) | MMRRC | ID# 015966-UCD | |
| Strain, strain background (*Mus musculus*) | Mouse: Tg(*Pcp2-Cre*) | The Jackson Laboratory | Stock# 004146 | |
| Cell line (*Homo sapiens*) | 293AAV | Cell Biolab | Cat#: AAV-100 | |
| Antibody | anti- GABAAR α1 (Rabbit polyclonal) | Frontier Inst. | Cat# GABAARa1-Rb-Af660 RRID:AB_2571571 | IHC(1 µg/ml) |
| Antibody | anti-GluD2 (Rabbit polyclonal) | Frontier Inst. | Cat# GluD2C-Rb-Af1200 RRID:AB_2571601 | IHC (1 µg/ml) |
| Antibody | anti-panAMPAR (Guinea pig polyclonal) | Frontier Inst. | Cat# panAMPAR-GP-Af580, RRID:AB_2571610 | IHC (1 µg/ml) IEM (5 µg/ml) |
| Antibody | anti-VGluT1 (Guinea pig polyclonal) | Frontier Inst. | Cat# VGluT1-GP-Af570, RRID:AB_2571534 | IHC (1 µg/ml) IEM (10 µg/ml) |
| Antibody | anti-VGluT2 (Guinea pig polyclonal) | Frontier Inst. | Cat# VGluT2-GP-Af810, RRID:AB_2341096 | IHC (1 µg/ml) |
| Antibody | anti-VIAAT (Guinea pig polyclonal) | Frontier Inst. | Cat# VGAT-GP-Af1000, RRID:AB_2571624 | IHC (1 µg/ml) |
| Antibody | anti-PSD95 (Guinea pig polyclonal) | Frontier Inst. | Cat# PSD95-GP-Af660, RRID:AB_2571539 | IHC (1 µg/ml) |
| Antibody | anti-GFP (Goat polyclonal) | Frontier Inst. | Cat# GFP-Go-Af1480, RRID:AB_2571574 | IHC (1 µg/ml) |
| Antibody | anti-calbindin (Goat polyclonal) | Frontier Inst. | Cat# Calbindin-Go-Af1040, RRID:AB_2532104 | IHC (1 µg/ml) |
| Antibody | anti-MAP2 (Goat polyclonal) | Frontier Inst. | Cat# MAP2-Go-Af860, RRID:AB_2571557 | IHC (1 µg/ml) |
| Antibody | anti-Car8 (Goat/Rabbit polyclonal) | Frontier Inst. | Cat# Car8-Go-Af780, RRID:AB_2571668 Cat# Car8-Rb-Af330, RRID:AB_2571667 | IHC (1 µg/ml) IEM (10 µg/ml) |
| Antibody | anti-Bassoon (Mouse monoclonal) | Enzo | Cat# SAP7F407, RRID:AB_2313990 | IHC (1 µg/ml) IEM (20 µg/ml) |
| Antibody | anti-GABAAR γ2 (Rabbit polyclonal) | Millipore | Cat#: AB5559, RRID: AB_11211236 | WB (1:2000) |

| Reagent type (species) or resource | Designation | Source or reference | Identifiers | Additional information |
|---|---|---|---|---|
| Antibody | anti-GARLH4 (Rabbit polyclonal) | *Yamasaki et al., 2017* | N/A | WB (0.1 µg/ml) |
| Antibody | anti-GluA1 (Rabbit polyclonal) | Millipore | Cat#: AB1504, RRID: AB_2113602 | WB (1:2000) |
| Recombinant DNA reagent | pAAV-DJ (plasmid) | Cell Biolabs | VPK-410-DJ | |
| Recombinant DNA reagent | pHelper (plasmid) | Cell Biolabs | VPK-410-DJ | |
| Recombinant DNA reagent | pAAV-MCS (plasmid) | Cell Biolabs | VPK-410 | |
| Recombinant DNA reagent | pAAV-Promoter less (plasmid) | This paper | N/A | pAAV-MCS (Cell Biolabs) |
| Recombinant DNA reagent | Nrxn3alpha pXY-Asc (plasmid) | Horizon | Cat# MMM1013-202798372 | |
| Sequence-based reagent | B1.Nrxn3a.For | This paper | PCR primers | GGGGACAAG TTTGTACAAAA AAGCAGGCTC CACCATGAG CTTTACCCT CCACTCAG TTTTCTTC |
| Sequence-based reagent | Nrxn3a(-).B5.Rev | This paper | PCR primers | GGGGACAACT TTTGTATACAA AGTTGTCACAT AATACTCCTTG TCCTTGTTTTT CTGTTTC |
| Sequence-based reagent | B5.(-M)AcGFP.Fo | This paper | PCR primers | GGGGACAA CTTTGTATA CAAAAGTT GTGAGCAA GGGCGCC GAGCTGTTC |
| Sequence-based reagent | AcGFP*.B2.Rev | This paper | PCR primers | GGGGACCAC TTTGTACAAGA AAGCTGGGT TCACTTGTAC AGCTCATCCATGCC |
| Sequence-based reagent | AsCI.DEST.for | This paper | PCR primers | TACATGGCGC GCCACAAGTT TGTACAAAAAAGC |
| Sequence-based reagent | DEST.BsiWI.rev | This paper | PCR primers | ATGTACGTAC GACCACTTTG TACAAGAAAGC |
| Sequence-based reagent | B5.cre.For | This paper | PCR primers | GGGGACAAC TTTGTATACAA AAGTTGCCACC ATGTCCAATTT ACTGACCG TACACC |
| Sequence-based reagent | cre*.B2.Rev | This paper | PCR primers | GGGGACCACT TTGTACAAGAA AGCTGGGTTCA ATCGCCATCTT CCAGCAGGCG |

| Reagent type (species) or resource | Designation | Source or reference | Identifiers | Additional information |
|---|---|---|---|---|
| Sequence-based reagent | B1.pNOS.For | This paper | PCR primers | GGGGACAAGT TTGTACAAAAA AGCAGGCTC CCCTCACCCAT CCCCACCCAC CTCCATCCATAC |
| Sequence-based reagent | pNOS.B5.Rev | This paper | PCR primers | GGGGACAA CTTTTGTAT ACAAAGTTGTT GCCGTTCGGC CTTGGGTGG CATGATTTC |
| Sequence-based reagent | B1.pGRIN3A.For | This paper | PCR primers | GGGGACAA GTTTGTACA AAAAAGCAG GCTCCCTGC CGTGCAAGGA CCACACATT CTACACTATAC |
| Sequence-based reagent | pGRIN3A.B5.Rev | This paper | PCR primers | GGGGACAACTT TTGTATACAAA GTTGTCGGCCA CCTTACCGCGG GCTCCCCCA GCGCCTGG |
| Sequence-based reagent | B1.pC-KIT.For | This paper | PCR primers | GGGGACAA GTTTGTACAAA AAAGCAGGCTG TCCACCCCCG GATAGCCACAG TGACTGTGAAATG |
| Sequence-based reagent | pC-KIT.B5.Rev | This paper | PCR primers | GGGGACAA CTTTTGTAT ACAAAGTTGT GTGCACCGAG CGCGGCAAA GCCGAGC |
| Commercial assay or kit | Endofree QIAGEN Maxi kit | QIAGEN | QIAGEN Cat#12,362 | |
| Chemical compound, drug | L-Glutamine | GIBCO | 25030–081 | |
| Chemical compound, drug | Penicillin– streptomycin | GIBCO | 15140–122 | |
| Chemical compound, drug | DMEM media | SIGMA | 11965092 | |
| Chemical compound, drug | Iscove's modified DM (IMDM) | Thermo | 12440053 | |
| Chemical compound, drug | Benzonase nuclease | Sigma | E1014-25kU | |
| Chemical compound, drug | Pfu Turbo DNA polymerase | Agilent | 600,250 | |
| Software, algorithm | MetaMorph | Molecular Devices | RRID:SCR_002368 | |

## Animals

All animal handling was in accordance with protocols approved by the Institutional Animal Care and Use Committee (IACUC) of Yale University (Animal Welfare Assurance# D16-00146, Animal protocol number 2021-11029), the Albert Einstein College of Medicine (Animal Welfare Assurance# A3312-011, Animal protocol number 00001043) and Hokkaido University, Japan (Approval number, #19-0111). Animal care and housing were provided by the Yale Animal Resource Center (YARC), in

compliance with the Guide for the Care and Use of Laboratory Animals (National Academy Press, Washington, DC, 1996). Wild-type (C57BL/6 J, Stock# 000664), the transgenic Cre mouse under the *Pcp2* promoter (*Barski et al., 2000*) (Stock# 004146) were obtained from the Jackson Laboratory. The transgenic Cre mouse under the *Gabra6* promoter (*Fünfschilling and Reichardt, 2002*) (ID# 015966-UCD) was obtained from MMRRC. *Pcp2-Cre: Stxbp1*$^{fl/fl}$ (ΔPC) mice exhibited mild ataxia and survived at least 6 months without extra care, whereas *Gabra6-Cre: Stxbp1*$^{fl/fl}$ (ΔGC) mice were unable to stand, requiring food pellets and water to be supplied on the cage floor for survival.

## Cell lines

293AAV cells were obtained directly from Cell Biolab, Inc and used them within 20 passages with continuous monitoring of cell morphology. Cells were grown at 37 °C, 5 % $CO_2$. The cell line was tested negative for mycoplasma contamination. The cell identity relies on Cell Biolabs (Cat#AAV-100).

## Antibodies

The following antibodies were used at the indicated concentrations: rabbit polyclonal antibodies to GABAAR α1 (rabbit, 1 µg/ml for IHC (immunohistochemistry), Frontier Inst. GABAARa1-Rb-Af660, RRID:AB_2571571), GFP (goat, 1 µg/ml for IHC, Frontier Inst. GFP-Go-Af1480, RRID:AB_2571574), panAMPAR (guinea pig, 1 µg/ml for IHC, 5 µg/ml for ImmunoEM, Frontier Inst. panAMPAR-GP-Af580, RRID:AB_2571610), GluD2 (rabbit, 1 µg/ml for IHC, Frontier Inst. GluD2C-Rb-Af1200, RRID:AB_2571601),VGluT1 (guinea pig, 1 µg/ml for IHC (*Miyazaki et al., 2003*), Frontier Inst. VGluT1-GP-Af570, RRID:AB_2571534),VGluT2 (guinea pig, 1 µg/ml for IHC (*Miyazaki et al., 2003*), Frontier Inst. VGluT2-GP-Af810, RRID:AB_2341096), VIAAT (guinea pig, 1 µg/ml for IHC (*Miyazaki et al., 2003*), Frontier Inst. VGAT-GP-Af1000, RRID:AB_2571624), PSD-95 (guinea pig, 1 µg/ml for IHC, 10 µg/ml for ImmunoEM (*Fukaya and Watanabe, 2000*), Frontier Inst. PSD95-GP-Af660, RRID:AB_2571539), calbindin (goat, 1 µg/ml for IHC, Frontier Inst. Calbindin-Go-Af1040, RRID:AB_2532104), Car8 (goat, 1 µg/ml for IHC; rabbit, 10 µg/ml for ImmunoEM (*Patrizi et al., 2008*), Frontier Inst. Car8-Go-Af780, Car8-Rb-Af330, RRID:AB_2571667/2571668), MAP2 (goat, 1 µg/ml for IHC, Frontier Inst. MAP2-Go-Af860, RRID:AB_2571557), Bassoon (mouse, 1 µg/ml for IHC, 20 µg/ml for ImmunoEM, Enzo, Cat# SAP7F407, RRID:AB_2313990), GABAARγ2 (rabbit, 1:2000 for immunoblot, Millipore. AB5559, RRID: AB_11211236), GARLH4 (rabbit, 0.1 µg/ml for immunoblot, *Yamasaki et al., 2017*), and GluA1 (rabbit, 1:2000 for immunoblot, Millipore, AB1504, RRID:AB_2113602).

## Immunostaining

Adult mice were deeply anesthetized with pentobarbital and perfused transcardially with 4 % para-formaldehyde in 0.1 M phosphate buffer (PB, pH 7.4). After postfixation for 3 hr, 50 µm sections were prepared using a vibratome (Leica). In staining for postsynaptic molecules, sections were incubated with 1 mg/ml pepsin (Dako) in 0.2 N HCl at 37 °C for 2 min to facilitate antibody access to antigen molecules condensed in the postsynapse (*Fukaya and Watanabe, 2000*), stained with the appropriate primary and secondary antibodies and mounted with ProLong Gold (Thermo). Data were acquired using confocal microscopy (Zeiss LSM 800) with Plan-APOCHROMAT ×63 oil immersion objective lens (Zeiss). Quantification analysis was performed using MetaMorph (Molecular Devices).

## Virus production

AAV was prepared as described previously (*McClure et al., 2011*; *Park et al., 2016*). Briefly, three plasmids encoding the required components for AAV production were transfected into 293 AAV cells (Cell Biolab, Inc) using the calcium phosphate methods: AAV-DJ, pHelper (Cell biolabs, Inc), and pAAV-CaM kinase II. For production of MLI-specific AAVs (pX), a putative promoter region of pAAV-synapsin vector was replaced into promoter region of *Nos-1*, *Grin3a*, *Kit* and obtained pX-GFP and pX-cre AAVs using with Gateway system (Thermo Fisher Scientific). After 48–60 hr post-transfection, cells were solubilized and AAVs were purified using a HiTrap Heparin column (GE healthcare).

## Stereotaxic AAV injection

Under sterile conditions, 3–4- week-old animals were anesthetized with isoflurane and secured in a stereotaxic frame. For cerebellar injection, holes for injecting needles were drilled into the occipital bone and coordinates were (0, 0, and 1.0 mm; caudal to the occipital external protuberance, right to

midline, and ventral to pial surface, respectively) with tilting the manipulator at 50° . For injection into ION, the needle was inserted into the medulla, and injections were done unilaterally and coordinates (1.0, 1.5, and 1.8 mm; rostral to the rostral tip of the occipital bone, right to midline, and ventral to pial surface, respectively), with tilting the manipulator at 50 degrees. All injections were done with 1 µl of AAV per region. Mice were analyzed at 4 weeks after injection.

## Electron microscopy

For conventional electron microscopy, mice were perfused transcardially with 2 % paraformaldehyde and 2 % glutaraldehyde in 0.1 M PB. The vibratome sections (400 µm in thickness) were immersed in 1 % OsO4 for 15 min, dehydrated by graded alcohol and embedded into Epon812. After polymerization at 60 °C for 48 hr, ultrathin sections (~80 nm) were prepared by ultramicrotome (UCT, Leica).

For postembedding immunogold electron microscopy, mice were perfused transcardially with 4 % paraformaldehyde in 0.1 M PB. As described previously (*Straub et al., 2016*), vibratome sections (400 µm in thickness) were cryoprotected with 30 % glycerol in PB, frozen rapidly with liquid propane in the EM CPC unit (Leica Microsystems), freeze-substituted to Lowicryl HM-20 resin (Electron Microscopy Sciences, Hatfield, PA), and polymerized with UV light by AFS unit (Leica). Ultrathin sections (~80 nm) on nickel grids were etched with saturation sodium-ethanolate solution for 1–5 s, and treated with following solutions: the donkey blocking solution containing 2 % normal donkey serum (Jackson ImmunoResearch) in Tris-buffered saline (TBS, pH 7.4) for 20 min, primary antibodies diluted with the donkey blocking solution overnight, and colloidal gold (10 nm)-conjugated species-specific IgG (1:100, British BioCell International, UK) in the donkey blocking solution for 2 hr. After extensive washing in TTBS and blocking with 2 % normal donkey serum, sections were further subjected to the second primary antibody and colloidal gold (15 nm)-conjugated species-specific secondary antibody. Sections were washed with TBS and distilled water, then stained with 1 % $OsO_4$ for 15 min, 2 % uranyl acetate for 4 min, and Reynold's lead citrate solution for 60 s. All electron microscopy images were taken with a JEM1400 electron microscope (JEOL, Japan). For quantitative analysis, postsynaptic membrane-associated immunogold particles, being defined as those apart <35 nm from the cell membrane, were counted on scanned electron micrographs and analyzed using MetaMorph software (Molecular Devices).

## Pull down with Fc fusion proteins

Either the extracellular domain of mouse neurexin-3 alpha SS4+ (Horizon, MMM1013-202798372) or the signal peptide from GluA1 was fused with the human Fc domain and cloned into pcDNA3 expression vector (Invitrogen). HEK293T cells (ATCC, CRL-3216) were transfected with the plasmids using Lipofectamine 2000 according to the manufacture's protocol. After 48 hr, the cultured media were collected and secreted Fc fusion proteins were purified with Protein A-sepharose (GE Healthcare). Mouse cerebella were solubilized with 25 mM Tris–Cl pH8.0, 1 % lauryl maltose neopentyl glycol (LMNG, Anatrace), Halt protease inhibitor (Thermo Fisher), and centrifuged at 100,000 × *g* for 30 min (*Yamasaki et al., 2017*). Supernatants were then incubated with Fc fusion proteins on the protein A beads with 2 mM $CaCl_2$ and 2 mM $MgCl_2$ or with 5 mM EDTA. Beads were then washed six times with 0.1 % LMNG, 40 mM Tris–HCl pH 8.0 with or without 2 mM $CaCl_2$, 2 mM $MgCl_2$. Bound proteins were eluted by heating the resin in 1× SDS–PAGE sample buffer and analyzed by SDS–PAGE. Input lanes contained 5 % of the protein used for immunoprecipitation.

## Two-photon imaging and MNI-glutamate uncaging

Acute sagittal cerebellar slices (250 µm thick) were prepared from adult control and ΔGC mice. Briefly, mice were anesthetized with isoflurane and euthanized by decapitation. Brains were removed and dissected using a VT1200S microslicer (Leica Microsystems Co.) in an ice-cold cutting solution containing (in mM): 110 choline, 2.5 KCl, 25 $NaHCO_3$, 1.25 $NaH_2PO_4$, 0.5 $CaCl_2$, 7 $MgCl_2$, 25 D-glucose, 11.6 sodium l-ascorbate, and 3.1 sodium pyruvate. Slices were then transferred and incubated for 15 min in a chamber placed in a warm bath (33–34°C) with oxygenated artificial cerebrospinal fluid (ACSF) solution containing (in mM): 124 NaCl, 2.5 KCl, 26 $NaHCO_3$, 1 $NaH_2PO_4$, 2.5 $CaCl_2$, 1.3 $MgSO_4$, and 10 D-glucose. Slices were then kept at room temperature for at least 30 min prior to recording. All solutions were equilibrated with 95 % $O_2$ and 5 % $CO_2$ (pH 7.4).

All electrophysiology and imaging experiments were performed at 32°C ± 1°C in a submersion-type recording chamber perfused at 2 ml/min with ACSF supplemented with the GABA$_A$ receptor antagonist picrotoxin (100 μM) and MNI-glutamate (2.5 mM). Whole-cell patch-clamp recordings using a Multiclamp 700 A amplifier (Molecular Devices) were made from PCs voltage clamped at −60 mV using patch-type pipette electrodes (~2–3 MΩ) containing (in mM): 135 KMeSO$_4$, 5 KCl, 1 CaCl$_2$, 5 NaOH, 10 HEPES, 5 MgATP, 0.4 Na$_3$GTP, 5 EGTA, and 10 D-glucose, pH 7.2. The morphological indicator Alexa 594 (20 μM) was added to the pipette and allowed to diffuse and equilibrate for >30 min.

An Ultima 2 P microscope (Bruker Corp.) equipped with an Insight DeepSee laser and a MaiTai HP laser (Spectra Physics) was used for imaging and uncaging. Excitation wavelength for imaging and uncaging were 820 and 720 nm, respectively. Uncaging laser was parked ~1 μm far from the head of the target spine, and two uncaging pulses (1 ms duration, 100 Hz) were elicited. Electrophysiological data were acquired at 5 kHz, filtered at 2.4 kHz using IgorPro 7.01 (Wavemetrics, Inc). Statistical significance was assessed using OriginPro (OriginLaboratory).

## Statistics

Quantification and statistical details of experiments can be found in the figure legends. All data are given as mean ± SEM. The normality of distributions was assessed using the Shapiro–Wilk test. In data sets that were normally distributed, paired or unpaired Student's $t$-test and one-way analysis of variance with Bonferroni post hoc test were used to assess between-group differences. For the nonparametric tests, Mann–Whitney $U$-test and Kruskal–Wallis test followed by Steel or Steel–Dwass post-test. Statistical significance was set to $p < 0.05$, and statistically significant differences are indicated as follows: $*p < 0.05$, $**p < 0.01$, and $***p < 0.001$.

## Acknowledgements

The authors thank Drs. Janghoo Lim, Pietro De Camilli, Shaul Yogev, and Marc Hammarlund for providing their valuable comments and the members of the Tomita lab for discussions. We thank Dr. Louis Reichardt for sharing transgenic Cre mice under the *Gabra6* promoter through the MMRRC; Dr. Michael Meyer for sharing transgenic Cre mice under the *Pcp2* promoter through the Jackson Laboratory; and Addgene for the plasmids listed in Experimental Procedures.

## Additional information

### Competing interests

The authors declare that no competing interests exist.

### Funding

| Funder | Grant reference number | Author |
|---|---|---|
| NIH Office of the Director | MH115705 | Susumu Tomita |
| NIH Office of the Director | MH077939 | Susumu Tomita |
| Ministry of Education, Culture, Sports, Science and Technology | Grant-in-Aid for Scientific Research 17K08485 | Taisuke Miyazaki |
| Ministry of Education, Culture, Sports, Science and Technology | Grant-in-Aid for Scientific Research 18K06813 | Taisuke Miyazaki |
| NIH Office of the Director | F32NS093952 | Yoav Noam |
| NIH Office of the Director | NS113600 | Pablo E Castillo |
| NIH Office of the Director | MH125772 | Pablo E Castillo |

| Funder | Grant reference number | Author |
|--------|-----------------------|--------|

The funders had no role in study design, data collection and interpretation, or the decision to submit the work for publication.

## Author contributions

Taisuke Miyazaki, Data curation, Formal analysis, Investigation, Methodology, Project administration, Validation, Visualization, Writing - original draft, Writing – review and editing; Megumi Morimoto-Tomita, Investigation, Methodology, Resources, Validation, Writing – review and editing; Coralie Berthoux, Formal analysis, Methodology, Writing – review and editing; Kotaro Konno, Investigation, Validation, Writing – review and editing; Yoav Noam, Data curation, Investigation, Methodology, Validation, Writing – review and editing; Tokiwa Yamasaki, Methodology, Writing – review and editing; Matthijs Verhage, Resources, Writing – review and editing; Pablo E Castillo, Data curation, Funding acquisition, Supervision, Writing – review and editing; Masahiko Watanabe, Data curation, Methodology, Supervision, Writing – review and editing; Susumu Tomita, Conceptualization, Data curation, Funding acquisition, Investigation, Methodology, Project administration, Resources, Supervision, Visualization, Writing - original draft, Writing – review and editing

## Author ORCIDs

Pablo E Castillo  http://orcid.org/0000-0002-9834-1801
Masahiko Watanabe  http://orcid.org/0000-0001-5037-7138
Susumu Tomita  http://orcid.org/0000-0001-8344-259X

## Ethics

All animal handling was in accordance with protocols approved by the Institutional Animal Care and Use Committee (IACUC) of Yale University (Animal Welfare Assurance# D16-00146, Animal protocol number 2021-11029), the Albert Einstein College of Medicine (Animal Welfare Assurance# A3312-011, Animal protocol number 00001043), and Hokkaido University, Japan (Approval number, #19-0111). Animal care and housing were provided by the Yale Animal Resource Center (YARC), in compliance with the Guide for the Care and Use of Laboratory Animals (National Academy Press, Washington, DC, 1996).

## Decision letter and Author response

Decision letter https://doi.org/10.7554/eLife.59613.sa1
Author response https://doi.org/10.7554/eLife.59613.sa2

# Additional files

## Supplementary files

• Transparent reporting form

## Data availability

All data generated or analyzed during this study are included in the manuscript and supporting file. Source Data files showing all raw values for each figure and the original images of uncropped blots for Figure 6B have been provided.

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
