## [Decision Letter]

**Acceptance summary:**

This paper by Miyazaki and colleagues investigates the consequences of cell elimination on retention of postsynaptic AMPA and GABAA receptors at denervated synapses. The authors find that loss of parallel fiber inputs to Purkinje cells does not lead to spine removal or loss of AMPARs. At GABAergic Purkinje cell to deep cerebellar nuclei synapses, loss of inputs results in loss of GABAARs.

**Decision letter after peer review:**

[Editors’ note: the authors submitted for reconsideration following the decision after peer review. What follows is the decision letter after the first round of review.]

Thank you for submitting your work entitled "Excitatory and inhibitory receptors utilize post-and trans-synaptic mechanisms in vivo" for consideration by *eLife*. Your article has been reviewed by 2 peer reviewers, and the evaluation has been overseen by a Reviewing Editor and a Senior Editor. The reviewers have opted to remain anonymous.

Our decision has been reached after consultation between the reviewers. Based on these discussions and the individual reviews below, we regret to inform you that your work will not be considered further for publication in *eLife*.

Although the authors appreciate your careful analyses, they raised several critical issues, including the generality of the findings and lack of electrophysiological data, as detailed in their comments attached below.

*Reviewer #1:*

This paper by Miyazaki and colleagues investigates the consequences of cell elimination on retention of postsynaptic AMPA and GABAA receptors at denervated synapses. Using conditional STXBP1 knockout mice to induce cell type-specific loss of neurons in the cerebellum, they find that loss of parallel fiber inputs to Purkinje cells does not lead to spine removal or loss of AMPARs, suggesting these receptors do not rely on presynaptic input for their postsynaptic localization. At GABAergic Purkinje cell to deep cerebellar nuclei synapses, loss of inputs results in loss of GABAARs, suggesting that these receptors require presynaptic input for their postsynaptic localization. Overexpression of the presynaptic adhesion molecule neurexin-3alpha in climbing fibers increases the presence of GABAARs at these synapses, suggesting that neurexin could be part of a mechanism that retains GABAARs at inhibitory postsynapses.

Overall the paper makes a number of interesting observations, but whether these can be generalized to other types of synapses remains unclear, as do the mechanisms involved.

1. The authors make the interesting observation that elimination of granule cells results in the appearance of dendro-dendritic synaptic contacts. Although AMPARs continue to localize at the PSD-like structures of these synapses as the authors conclude, the situation here is distinct from the PF-PC synapse where elimination of granule cells denervates spines. This makes it difficult to draw a general conclusion here.

2. The authors propose neurexin-3 as a putative interactor of GABAARs and use overexpression to analyze effects on postsynaptic GABAAR retention. It is now clear on what basis neurexin-3alpha is proposed as putatibe interactor; to my knowledge the role of neurexin-3alpha in GABAAR interaction has not been reported. This overexpression experiment does not prove that neurexins retain GABAARs postsynaptically. Furthermore, neurexin-3 regulates postsynaptic AMPAR levels (Aoto et al., Cell 2013), which seems at odds with the authors' previous point that AMPARs do not require presynaptic input to localize at the PSD. In addition, the study by Zhang et al., Neuron 2010 that showed that neurexins can directly interact with GABAARs should be cited here.

3. The specific splice variant of neurexin-3 used in this study should be indicated; I was unable to find this information.

4. In the Discussion it is mentioned that 'free spines' in denervated Purkinje cells are 'unusual types of structures', however free spines have been observed in work from the Yuzaki lab on Cbln1 and GluD2 knockout mice, which should be acknowledged.

5. The use of the STXBP1 conditional knockout mouse to eliminate neurons should be better discussed.

*Reviewer #2:*

This manuscript by Miyazaki and colleagues attempted to address an interesting question- how postsynaptic receptors are localized in vivo. The authors took advantage of using various Cre-driver lines, specifically active in specific cell-types of cerebellum, crossed with STXBP1 floxed line.

Most imaging data look convincing and support the conclusion of each data, but paper does not clearly read. I suppose that part of this was likely due to the fact that the manuscript was submitted as a short report. In addition, I am not sure whether observations made in cerebellar circuits could be universally applied to other neural circuits in other brain areas, as opposed to the title of this manuscript. Some of underlying logics (e.g. why STXBP1 homozygous neurons were used in addressing the question, and why Nrxn3alpha-GFP signals were analyzed in Figure 3 etc) were not clearly explained and/or justified in texts. Most importantly, lack of electrophysiology in this manuscript significantly lessens my enthusiasm to support the publication of the current manuscript in e*Life*.

---

## [Author Response]

[Editors’ note: The authors appealed the original decision. What follows is the authors’ response to the first round of review.]

Reviewer #1:[…] Overall the paper makes a number of interesting observations, but whether these can be generalized to other types of synapses remains unclear, as do the mechanisms involved.1. The authors make the interesting observation that elimination of granule cells results in the appearance of dendro-dendritic synaptic contacts. Although AMPARs continue to localize at the PSD-like structures of these synapses as the authors conclude, the situation here is distinct from the PF-PC synapse where elimination of granule cells denervates spines. This makes it difficult to draw a general conclusion here.

We would like to clarify the case of the PF-MLI synapse now reported in new Figure 3 (previous Figures 2F-I). In wild type mice, parallel fibers (PFs) form synapses with molecular layer interneurons (MLI). Following the elimination of cerebellar granule cells, no PF-MLI synapses could be detected. Instead, we found that most PSDs were found in both sides of dendro-dendritic contacts between MLIs (Figure 3B). We think these contacts do not represent synapses by three reasons:

1. We did not observe synaptic vesicles about 40 nm in diameter nearby the contact (Figure 3B).

2. As we described in the main text (page 6, first paragraph), “the cleft width of the ectopic contact (30.0 ± 1.0 nm) was wider than that of the synaptic cleft in control mice (15.8 ± 0.5 nm) (*n* = 63 – 73 from three mice each)”, which is more consistent with an adhesive junction than a synaptic junction.

3. Our new results (Figures 3D, E, F) show absence of Bassoon and VGluT1 at the dendro-dendritic contact sites, suggesting absence of the major components of excitatory presynaptic terminals (Figure 3G).

The lack of presynaptic machinery and the wide cleft of these ectopic dendro-dendritic contacts strongly suggest these contacts are non-synaptic. Thus, our conclusion that AMPAR clustering is maintained after presynaptic neuron ablation is supported by two distinct, but converging examples of PSD-like specializations on dendritic spines (PCs) and dendritic shafts (MLIs). We have clarified these points in the main text (page 6, 1^st^ paragraph).

2. The authors propose neurexin-3 as a putative interactor of GABAARs and use overexpression to analyze effects on postsynaptic GABAAR retention. It is now clear on what basis neurexin-3alpha is proposed as putatibe interactor; to my knowledge the role of neurexin-3alpha in GABAAR interaction has not been reported. This overexpression experiment does not prove that neurexins retain GABAARs postsynaptically. Furthermore, neurexin-3 regulates postsynaptic AMPAR levels (Aoto et al., Cell 2013), which seems at odds with the authors' previous point that AMPARs do not require presynaptic input to localize at the PSD. In addition, the study by Zhang et al., Neuron 2010 that showed that neurexins can directly interact with GABAARs should be cited here.

We thank the reviewer for his/her critical assessment of our findings. The reviewer raises three points:

To my knowledge the role of neurexin-3alpha in GABAAR interaction has not been reported.

This is correct. An interaction of Nrxn3alpha with GABAAR has not been shown. Thus, we now provided new data showing that the extracellular domain of neurexin-3alpha fused with Fc pulled down NL2, GABA_A_Rγ2 subunits, GARLH4, but not GluA1 AMPAR, from adult mouse cerebellar lysate in a calcium dependent manner (new Figure 6B).

Furthermore, neurexin-3 regulates postsynaptic AMPAR levels (Aoto et al., Cell 2013), which seems at odds with the authors' previous point that AMPARs do not require presynaptic input to localize at the PSD.

Aoto et al., Cell 2013 showed constitutive knockin of SS4+ in neurexin 3 showed changes in AMPARs and conditional switch from SS4+ to SS4- rescues AMPAR phenotypes. Since this is a constitutive knockin animal, the phenotypes may be induced during development. In contrast, we eliminated presynaptic input after development. Thus, we believe our data are not inconsistent with the Aoto et al. report.

In addition, the study by Zhang et al., Neuron 2010 that showed that neurexins can directly interact with GABAARs should be cited here.

The paper is now cited (page 8, 1^st^ paragraph).

3. The specific splice variant of neurexin-3 used in this study should be indicated; I was unable to find this information.

We used mouse neurexin3-alpha with SS4+. We have now clarified this point (page 14, last paragraph).

4. In the Discussion it is mentioned that 'free spines' in denervated Purkinje cells are 'unusual types of structures', however free spines have been observed in work from the Yuzaki lab on Cbln1 and GluD2 knockout mice, which should be acknowledged.

We meant to say unusual types of structures “in wild type mice”, since free spines are extremely rare in wild type mice. We have clarified this point and acknowledged the papers from the Yuzaki lab in the revised manuscript (page 10, last paragraph).

5. The use of the STXBP1 conditional knockout mouse to eliminate neurons should be better discussed.

Thanks for the suggestion. In response, we have now expanded the discussion to address these points clearly (page 9, 1^st^ paragraph).

Reviewer #2:This manuscript by Miyazaki and colleagues attempted to address an interesting question- how postsynaptic receptors are localized in vivo. The authors took advantage of using various Cre-driver lines, specifically active in specific cell-types of cerebellum, crossed with STXBP1 floxed line.Most imaging data look convincing and support the conclusion of each data, but paper does not clearly read. I suppose that part of this was likely due to the fact that the manuscript was submitted as a short report.

We thank the reviewer for his/her comments regarding the quality of our data. We also apologize for the poor readability of our manuscript. We totally agree with the reviewer that the short format did not help us and, precisely for this reason, we have revised our manuscript as a full article.

In addition, I am not sure whether observations made in cerebellar circuits could be universally applied to other neural circuits in other brain areas, as opposed to the title of this manuscript.

We are sorry for the confusion. We did not mean to say that all circuits in the brain utilize the same mechanisms that we describe in the cerebellum, a brain structure that offers significant advantages for the manipulation of identified presynaptic and postsynaptic neurons. To our knowledge, our study is the first one to demonstrate changes in receptor maintenance at postsynapses in the absence of presynaptic terminals in the mature brain, while analyzing four distinct synapses in the same brain structure: PF-PC and PF-MLI excitatory synapses, and PC-DCN and MLI-PC inhibitory synapses.

We acknowledge we failed to provide a good justification for the use of the cerebellum as a model system. It is well known that postsynapses without presynaptic terminals are quite unstable and rarely seen outside the cerebellum (Yuste and Bonhoeffer, 2004). Regardless, and in response to the reviewer’s comment, we utilized two different methods to eliminate presynaptic neurons in the hippocampus. 1, AAV-Cre injection into CA3 of *Stxbp1* floxed mice, which eliminates neurons cell-autonomously. 2, Coinjection of AAV-Cre and AAV-Flex-DTA into CA3 of wild type mice, which kills neurons by DTA, although DTA released from dead neurons may affect surrounding neurons. Using these two systems, we confirmed the elimination of presynaptic neurons (Author response image 1). However, unlike cerebellar Purkinje cells, we observed only 0.6% of free spines. These results strongly suggest that free spines are unstable in hippocampus, which is consistent with no free spines in hippocampus (Yuste and Bonhoeffer, 2004). We have now described these points in the discussion of our manuscript (page 10, last paragraph), but did not include these results in order to keep the focus of our manuscript. Nevertheless, if necessary, we will be happy to include this figure in the manuscript.

**Author response image 1. sa2fig1:** Elimination of presynaptic neurons reduces synapse number without producing free spines in hippocampus. (A–C) Electron micrographs showing CA1 region of wild-type (A) and *Stxbp1*^fl/fl^ mice injected with AAV-synapsin promoter-driven (pSynapsin) Cre (B) and wild-type injected with AAV-pSynapsin Cre and -pSynapsin flex diphteria toxin A (DTA). AAVs were injected at both sides of CA3 region at P30–40. In CA3 elimination groups (B, C), spines contacting to terminals with forming asymmetry synapse (asterisks) was significantly decreased. (D, E) Graphs showing the number of synapses at CA1 regions (D) (N/100 µm^2^, *n* = 10–30 images from three mice) and the ratio of free spine relative to total spines (E) (*n* = 10–30 images from three mice). Note free spines were rarely produced in elimination of CA3 neuron. Data are shown as means ± SEMs, Kruskal–Wallis test followed by Steel–Dwass post-test; ***p < 0.001.

Some of underlying logics (e.g. why STXBP1 homozygous neurons were used in addressing the question, and why Nrxn3alpha-GFP signals were analyzed in Figure 3 etc) were not clearly explained and/or justified in texts. Most importantly, lack of electrophysiology in this manuscript significantly lessens my enthusiam to support the publication of the current manuscript in eLife.

By changing the format of the manuscript to a full article, we now provide a better rationale and justification for our experiments, which are highlighted by red color. Briefly, while a typical killer cell approach relies on the use of toxins, the generated dead neurons may release such toxins and affect surrounding cells. In contrast, the Stxbp1 approach requires homozygous mutations for cell death and provides a cell autonomous system to eliminate cells. The Nrxn3a-GFP approach is now fully explained in the text (page 3, 2^nd^ paragraph and page 9, 1^st^ paragraph).

Finally, the revised manuscript now includes new electrophysiology data that address a key question, namely, whether AMPARs on free spines are functional. We found that uncaged glutamate-evoked responses (uEPSCs) from dendritic spines in wild type mice and free spines in ΔGC mice (Figure 2E-H) are virtually identical. These new findings demonstrate that the remaining AMPARs are functional.

Reference:

Yuste, R., and Bonhoeffer, T. (2004). Genesis of dendritic spines: insights from ultrastructural and imaging studies. Nat Rev Neurosci 5, 24-34.